# Analysis of Spatial Patterns and Socioeconomic Activities of Urbanized Rural Areas in Fujian Province, China

Qinghai Guo [1,2], Zhichao He [3,4,*], Dawei Li [5] and Marcin Spyra [3,6]

1   School of Civil Engineering and Architecture, Zhejiang Sci-Tech University, Hangzhou 310018, China; qhguo@zstu.edu.cn
2   Zhejiang Academy of Ecological Civilization, Hangzhou 310018, China
3   Department of Sustainable Landscape Development, Institute for Geosciences and Geography, Martin-Luther-University Halle-Wittenberg, 06120 Halle (Saale), Germany; marcin.spyra@geo.uni-halle.de
4   Land Change Science Research Unit, Swiss Federal Research Institute WSL, 8903 Birmensdorf, Switzerland
5   School of Geography and Planning, Sun Yat-sen University, Guangzhou 510006, China; lidawei3@mail.sysu.edu.cn
6   Faculty of Civil Engineering and Architecture, Opole University of Technology, 45758 Opole, Poland
*   Correspondence: zhichao.he@student.uni-halle.de

**Abstract:** Rural urbanization under China's process of rapid urbanization entails significant rural transformation and is profoundly influencing sustainable development. However, little research has been undertaken on spatial patterns and socioeconomic activities. In this study, we defined urbanized rural areas as territories where the population size, economic output, and built-up land area are larger than in other rural and urban areas. Using large-scale and high-granularity spatial data, we delimited 255 urbanized rural areas from the 15,117 village-level administrative units in Fujian Province, China, in 2015. Analysis of the spatial patterns of the urbanized rural areas showed that spatial clustering, proximity to well-developed urban centers, and transportation accessibility influenced the development of the urbanized rural areas. Analysis of socioeconomic activities in the urbanized rural areas showed that the urbanized rural areas are rudimentary urban areas in terms of socioeconomic activities. Specifically, we found four representative socioeconomic activities in the urbanized rural areas: an urban-like housing model, diverse non-agricultural activities, transportation improvements, and sufficient health services. Based on our findings, we put forward several policy implications. This study can add valuable new knowledge for rural and urbanization studies.

**Keywords:** rural urbanization; delimitation; spatial patterns; non-agricultural economy; POI; sustainable development; China

## 1. Introduction

The impacts of rural transformation on the implementation of the United Nations' Sustainable Development Goals (SDGs) are not inferior to those of urbanization [1–4]. The SDGs aim for the relocation of rural areas from the margin of our attention to the foreground. Rural areas, especially in developing countries, are areas where most of the inhabitants struggle with poverty (SDG 1) [5], hunger (SDG 2) [6], insufficient health care (SDG 3) [7], and poor education (SDG 4) [8]; where the contribution of agricultural sectors to the economy is shrinking, with these sectors providing fewer job opportunities than before (SDG 8) [1]; and where ecosystems are particularly vulnerable to climate change (SDGs 13 and 15) [4]. Thus, more research into and more policies addressing rural transformation in developing countries could contribute to more effective implementation of the SDGs.

Traditionally, rural areas differ markedly from urban areas in terms of their small population size and economic output, agriculture-dominated economic activities, and natural landscape. Rural urbanization is a process in which rural areas become urbanized, such as in desakotas [9], town villages [10,11], hidden urbanization [12], in situ urbanization [13],

and urbanized villages [14] (Table 1). Urbanized rural areas are often located around large- and medium-sized cities, where the suburbanization process transforms rural areas to urbanized rural areas [15–17]. Thus, urbanized rural areas are neither urban areas nor rural areas, but instead demonstrate features of both because the demographic, economic, and environmental spaces in these areas have transformed into more complex ones than ever before [18,19].

**Table 1.** Comparison of studies on rural urbanization.

| | Definition | Spatial Patterns | Socioeconomic Activities |
|---|---|---|---|
| Desakotas | Regions with anintense mixture of agricultural and nonagricultural activities that often stretch along corridors between large city cores | Desakotas are frequently characterized by well-developed road and canal infrastructures that allow intense movement of commodities and people | The types of nonagricultural economic activities in desakotas are diverse and include trading, transportation, and industry |
| Urbanized villages | Rural villages with typical complex and multifunctional suburban landscapes | One urbanized village is localized approximately 5 km west of the city of Ghent. The urbanized village was disclosed early by the railway and is situated on a main access road to the city | More new residential, commercial, and industrial development |
| Town villages | Town villages are villages with considerable development of the nonagricultural economy | None | People in town villages are called farmers but engage in nonagricultural economic activities, such as small manufacturing, agricultural byproduct processing, trading, catering, tourism, and transportation |
| Hidden urbanization | Hidden urbanization refers to rural settlements with economic profiles firmly based in the secondary and tertiary sectors of the economy and with very high population densities | Hidden urbanization settlements are predominantly located right next to or in the proximity of secondary cities | New construction can be observed throughout all these urbanizing villages. Livelihoods in the urbanizing villages are based on the secondary and tertiary economic sectors, but for a number of households farming is still the most important part of their livelihood strategy and lifestyle |
| In situ urbanization | In situ urbanization refers to quasi-urban settlements with high population densities, where the majority of the workforce engages in non-agricultural activities | In situ urbanization demonstrates a relative concentration of location | Most farmers work in township and village enterprises |

Drawing on the literature, we can summarize three peculiarities of urbanized rural areas in terms of their demography, economy, and landscape. Demographically, urbanized rural areas have large population sizes as a result of their population mobility changing from unidirectional out-migration to bidirectional urban–rural migration [20]. Some researchers have found that the proportions of migrants flocking into rural areas and of villagers continuing to reside in their villages have been increasing [13,21,22]. Urbanized rural areas have high economic outputs. Villagers in urbanized rural areas remain farmers, but they engage in various non-agricultural activities to increase their incomes. For example, the villagers in the Kaliabu village in Indonesia could increase their incomes by providing online logo design services to international customers [23]. Other non-agricultural economic activities, such as tourism [24–26], E-commerce [27,28], and rural enterprises [13,29,30],



have also been demonstrated as effective approaches to increasing the rural economy. Regarding the landscape, while urbanized rural areas maintain their native rural landscapes, a considerable amount of land is often transformed into built-up land to accommodate non-agricultural activities, such as housing, transportation, and industry [31]. Based on the above literature, we define urbanized rural areas as rural areas with a larger population, economy, and built-up land area than other rural areas and urban areas.

The diverse transformations related to the demography, economy, and landscape in urbanized rural areas may have diverse impacts on sustainable development. For example, the various non-agricultural activities in rural areas markedly increase incomes and alleviate poverty, which contributes to SDG 1. Rural urbanization can also improve the accessibility of urban services to rural residents via improved transportation (related to SDG 9) [32,33]. However, the other SDGs, including mitigating land degradation, biodiversity loss, and climate change, might be very difficult to achieve because of built-up land expansion in urbanized rural areas. Rural heritage may be threatened due to rapid urbanization and economic growth in urbanized rural areas [34]. Urbanized rural areas may also lose out to gentrification, in which the satisfaction of the interests of one group comes from costs for another group [35]. Thus, in urbanized rural areas, these different concerns have to be taken seriously for landscape management and planning procedures [36].

While researchers and policy-makers are paying increasing attention to rural urbanization, most research has been focused on one or several sites, making it difficult to validate findings and to generalize over other sites. In this study, we addressed three research questions:

1. Are urbanized rural areas a fairly prevalent phenomenon or are there just isolated cases?
2. What are the spatial patterns of the urbanized rural areas that we delimited in Fujian Province?
3. What are the socioeconomic activities in the urbanized rural areas that we delimited in Fujian Province?

## 2. Materials and Methods

### 2.1. Study Area

We selected Fujian Province, China, as the study area (Figure 1) because its rural areas are experiencing both rural marginalization and rural urbanization. The rural population in Fujian Province decreased from 19.78 million in 2000 to 13.48 million in 2018. During the same period, the urban population increased from 14.32 million to 25.93 million. Rural depopulation is common in the western mountainous and hilly areas due to rapid urbanization, high elevation, and remoteness, resulting in rural marginalization [37]. However, some rural areas, mainly those concentrated in the eastern coastal regions, are urbanized, with villagers transforming their settlements into urban-like ones and obtaining jobs without having to migrate to urban areas [10,13,38]. Villagers' incomes in these areas have increased significantly. In addition, urbanized rural areas have been the main source of built-up land expansion in Fujian Province [39]. Fujian Province thus provides an ideal laboratory to explore our research questions. The 15,117 village-level administrative units in Fujian Province serve as research units delimiting the urbanized rural areas. These units are legalized grassroots governance units with defined boundaries where a villagers' committee is elected as the authority. Thus, they represent the basic socioeconomic units in rural China (e.g., for the census, the postal system, and for land ownership).

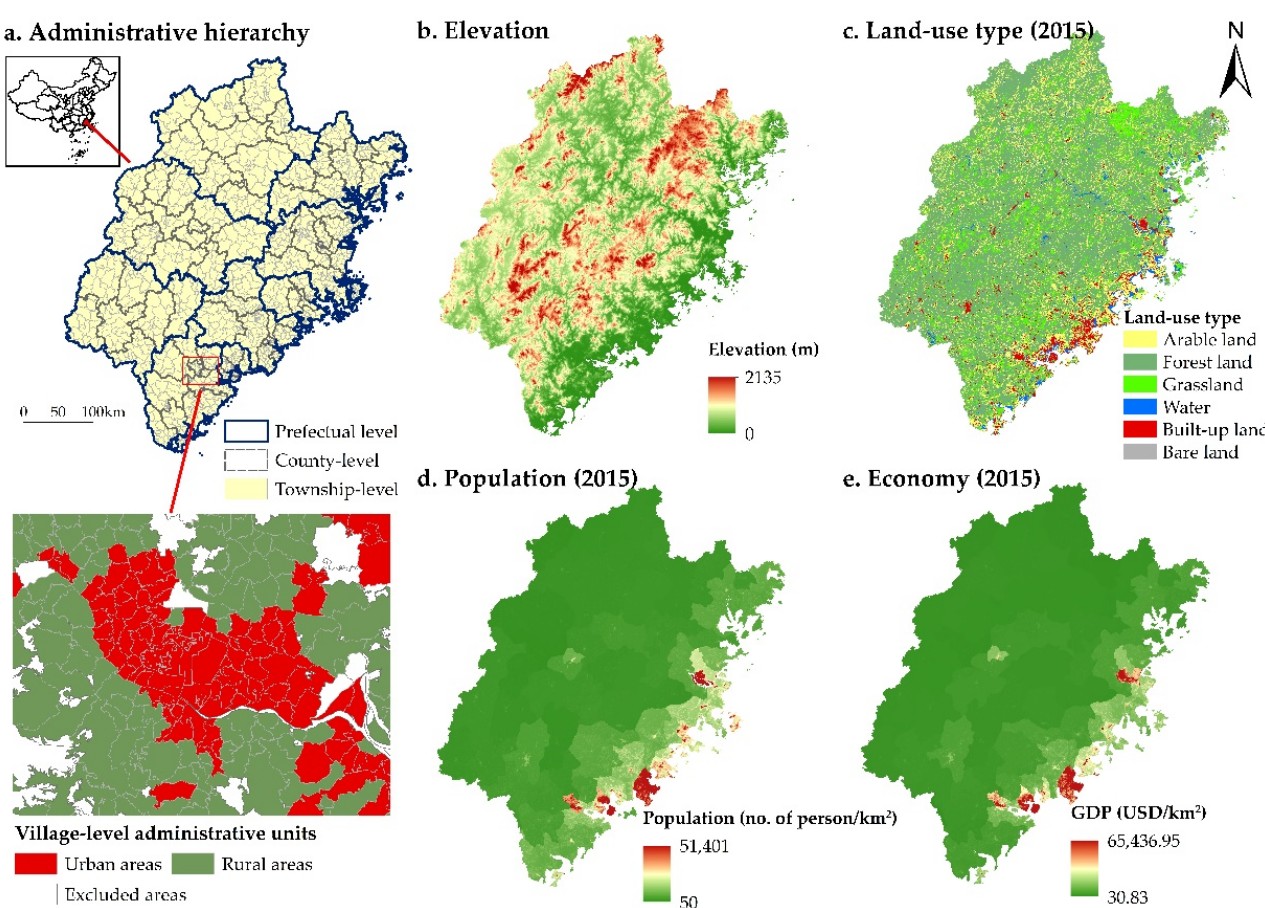

**Figure 1.** Fujian Province study area, China. (**a**) Administrative hierarchy: the administrative hierarchy in China is divided into five levels: provincial, prefectural, county, township, and village. Fujian Province has 9 prefecture-level cities, 84 counties, 1263 towns, and 15,117 village-level administrative units. (**b**) Elevation: the topography of Fujian Province is dominated by mountains and hills. (**c**) Land-use types in 2015: arable land, forest land, grassland, water, built-up land, and bare land. (**d**) Population and (**e**) economy in 2015: population and economic development are mainly concentrated in the eastern coastal regions.

*2.2. Data Sources*

The data on the boundaries of the 15,117 village-level administrative units in Fujian Province came from the local government. The data on the population, gross domestic product (GDP), and land use in 2015 came from the Data Center for Resources and Environmental Sciences, Chinese Academy of Sciences (http://www.resdc.cn). The data on the population and GDP had a raster format with a spatial resolution of 1 × 1 km (Figure 1d,e). The raster data were interpolated based on the economic and demographic census data, nighttime light data, and settlement distribution data. The land-use data had a vector format and were manually interpreted and produced from Landsat TM and Landsat 8 imagery with six land-use types (i.e., arable land, forest land, grassland, water, built-up land, and bare land) (Figure 1c). Built-up land corresponded to urban land, rural residential land, and other built-up land (e.g., mining, industrial, or transportation areas).

We collected the POI data from NavInfo (http://www.navinfo.com/en/index.aspx, accessed on 15 February 2022), which is the largest digital map provider in China. Each POI indicates the longitude, latitude, name, and the three levels of categories. There are 20 first-level categories, 185 second-level categories, and 578 third-level categories in the POI data. We used the third-level POI categories, which enabled us to investigate detailed socioeconomic activities.

*2.3. Method*

Our method consisted of three steps: (1) delimitation of the urbanized rural areas, (2) analysis of the spatial patterns of the urbanized rural areas, and (3) analysis of socioeconomic activities in the urbanized rural areas (Figure 2). Each step facilitated the answering of one of the research questions described above.

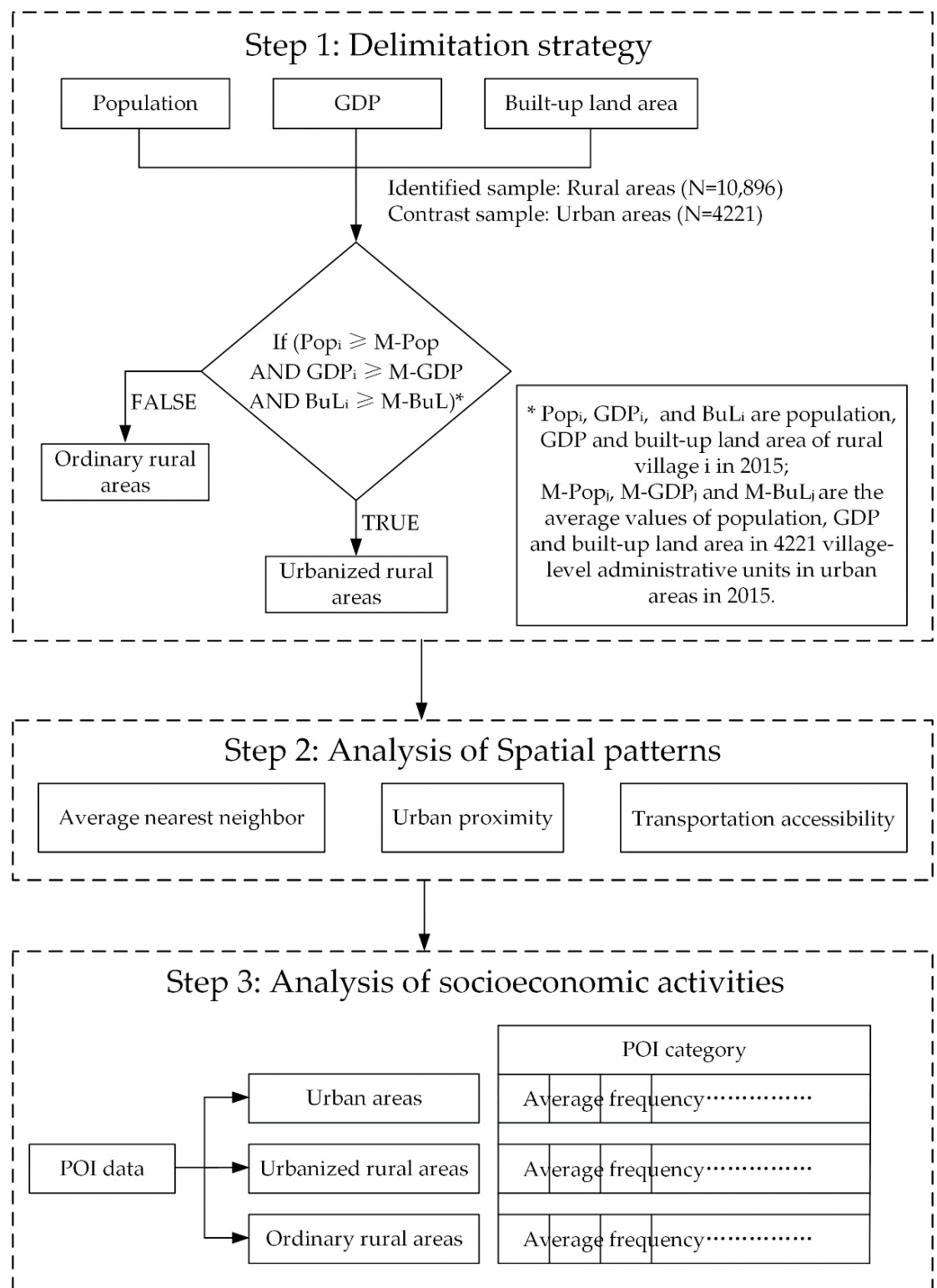

**Figure 2.** Three-step research process used in this study.

2.3.1. Delimitation Strategy

In this study, we used three indicators to delimit urbanized rural areas: population, GDP, and built-up land area. We used the 4221 village-level administrative units in urban areas as contrast samples and the average values of the three indicators as the standard of delimitation. Specifically, for a village-level administrative unit to be classified as an urbanized rural area, its population, GDP, and built-up land area all had to be greater than the corresponding average values for urban areas. If any one of the three indicators was not satisfied, the village-level administrative unit was classified as an ordinary rural area.

We extracted the population, GDP, and built-up land area of each village-level administrative unit, using the Spatial Statistics tools in ArcGIS 10.6. Next, we categorized 10,896 village-level administrative units as rural areas and 4221 village-level administrative units as urban areas based on an urban–rural code. The code was published by the National Bureau of Statistics of the People's Republic of China and is used for census.

2.3.2. Analysis of Spatial Patterns

We analyzed three aspects of the spatial patterns of the urbanized rural areas in Fujian Province: (1) the average nearest neighbor ratio; (2) urban proximity; and (3) transportation accessibility. For the average nearest neighbor ratio, we divided the average distance from each urbanized rural area to its nearest neighbors by the expected average distance assuming the urbanized rural areas were randomly distributed. If the ratio was lower than 1, the urbanized rural areas were considered clustered at the provincial level, and if the ratio was greater than 1, their distribution was considered dispersed. We calculated the average nearest neighbor ratio using the Average Nearest Neighbor tool in ArcGIS 10.6. Urban proximity was defined as the Euclidean distance from the border of an urbanized rural area to its nearest city center, which was calculated using the Near tool in ArcGIS 10.6. The city centers refer to the government seats of the nine prefectural cities in Fujian Province. Transportation accessibility was defined as the Euclidean distance from the border of an urbanized rural area to the national highways, which was again calculated using the Near tool in ArcGIS 10.6.

2.3.3. Analysis of Socioeconomic Activities

To answer the third research question concerning the socioeconomic activities in urbanized rural areas, we used the POI data. The original function of the POI data is to mark a user's location or destination in digital maps. The accuracy and efficiency of POI data in measuring socioeconomic activities have been demonstrated [40–42]. We first calculated the frequency of each third-level POI category within the urbanized rural areas, ordinary rural areas, and urban areas. We then divided the POIs' frequencies by the numbers of urbanized rural areas, ordinary rural areas, and urban areas. Finally, we selected the top 25 POI categories according to the average frequency.

**3. Results**

*3.1. Spatial Patterns of the Urbanized Rural Areas in Fujian Province*

Table 2 shows the average values for population, GDP, and built-up land area for the village-level administrative units in urban areas, urbanized rural areas, and ordinary rural areas. As defined here, urbanized rural areas have larger population, GDP, and built-up land area than urban areas. We delimited 255 urbanized rural areas in Fujian Province in 2015. These only account for a few of the rural areas. Among the nine prefecture-level cities, Quanzhou City possessed the largest number of urbanized rural areas (146). Second in rank was Zhangzhou City (59). These two cities are the hotspots of urbanization and private economic development in Fujian Province. Xiamen, Fuzhou, and Longyan City have 18, 13, and 10 urbanized rural areas, respectively. The remaining cities (Putian, Nanping, Sanming, and Ningde City) had very few urbanized rural areas (Figure 3).

**Table 2.** Average values for population, GDP, and built-up land area in urban areas, urbanized rural areas, and ordinary rural areas in Fujian Province in 2015.

| Regions | Population (No. of People) | GDP (USD) | Built-Up Land Area (km²) | No. of Villages |
|---|---|---|---|---|
| Urban areas | 3709 | 43,895,573.96 | 0.61 | 4221 |
| Urbanized rural areas | 7771 | 85,026,776.67 | 1.23 | 255 |
| Ordinary rural area | 1966 | 17,121,522.61 | 0.16 | 10,641 |

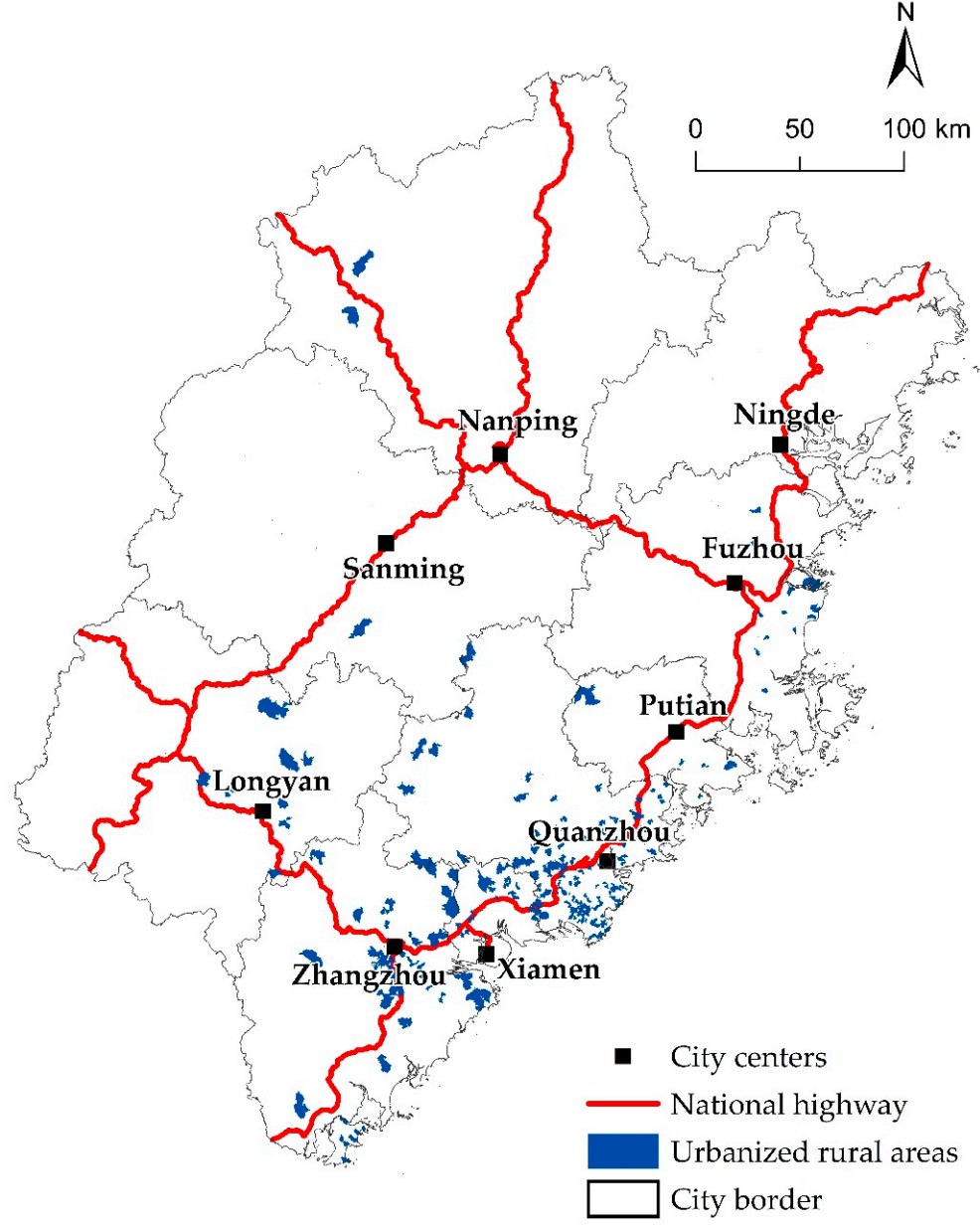

**Figure 3.** The spatial distribution of 255 urbanized rural areas in nine prefecture-level cities in Fujian Province in 2015.

The average nearest neighbor ratio was 0.50 (z-score = −15.41, *p*-value < 0.01), demonstrating that the urbanized rural areas were spatially clustered in Fujian Province. Approximately 90% of the urbanized rural areas were concentrated within 40 km of city centers and within 25 km of national highways (Figure 4).

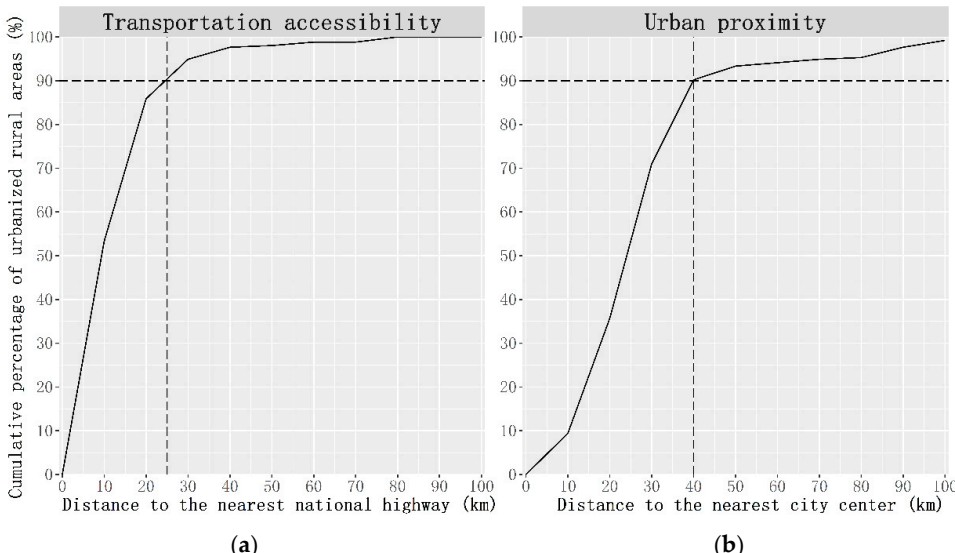

(**a**)  (**b**)

**Figure 4.** Relationships between transportation accessibility (**a**), urban proximity (**b**), and urbanized rural areas in 2015.

### 3.2. Socioeconomic Activities in the Urbanized Rural Areas in Fujian Province

In 2015, 699,267 POIs were distributed in 4221 urban areas, 13,505 POIs were distributed in 255 urbanized rural areas, and 134,380 POIs were distributed in 10,641 ordinary rural areas. On average, there were 165.66, 52.96, and 12.63 POIs for each of the village-level administrative units in the urban areas, urbanized rural areas, and ordinary rural areas, respectively. The urbanized rural areas and urban areas were similar in terms of the top 25 POI categories (Figure 5).

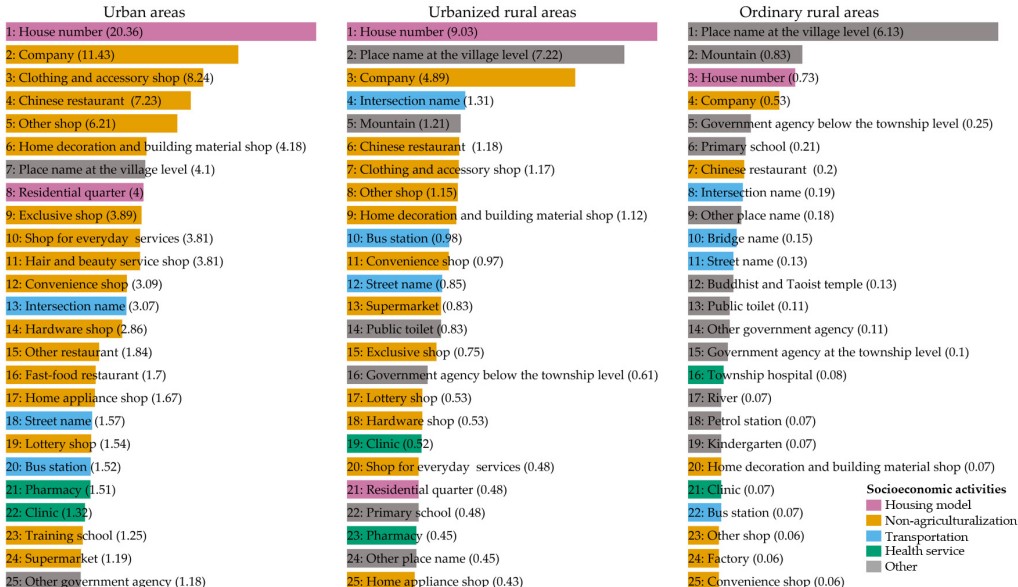

**Figure 5.** Average frequencies of the top 25 POI categories in the urbanized rural areas, ordinary rural areas, and urban areas in 2015. The top 25 POI categories were divided into: housing model, non-agriculturalization, transportation, health services, and other. The numbers in parentheses represent the average frequencies of the POI categories. For example, the average frequency of house number POIs in the urban areas was 20.36, which indicates that each village-level administrative unit in the urban areas had 20.36 house number POIs. Descriptions of the top 25 POI categories are given in Table S1.

The POI house number data ranked first in the urban areas and urbanized rural areas. The urban areas demonstrated the highest average frequency for house number POIs (20.36). The urbanized rural areas had 9.03 house number POIs, while the ordinary rural areas only had 0.73. In addition, the urban areas had four residential quarter POIs in each village-level administrative unit. The residential quarter POI data ranked 21st in the urbanized rural areas (average frequency = 0.48) and did not appear in the top 25 POI categories in the ordinary rural areas.

The top 25 POI categories relating to non-agricultural activities were very common in the urbanized rural areas, and their average frequencies were significantly higher than in the ordinary rural areas. For example, the average frequency of company POIs in the urbanized rural areas was 4.89, which was considerably higher than the value of 0.53 for the ordinary rural areas. The presence of POI categories relating to non-agricultural activities indicates that villagers in the urbanized rural areas are able to engage in economic activities beyond agriculture, such as being an employee (company POI), selling clothes (clothing and accessory shop POI), and catering (Chinese restaurant POI), and greatly increase their incomes locally.

The intersection name, street name, and bus station POIs indicate transportation improvements in the urbanized rural areas. The average frequencies of the intersection name, bus station, and street name POIs in the urbanized rural areas were 1.31, 0.98, and 0.85, respectively, which were higher than those in the ordinary rural areas.

Significant improvements in health services were evident in the urbanized rural areas. The urbanized rural areas had an average of 0.52 clinic POIs (ranked 19th) and 0.45 pharmacy POIs (ranked 23rd) in each village-level administrative unit. In the ordinary rural areas, the average frequencies of the township hospital and clinic POIs were only 0.08 (ranked 16th) and 0.07 (ranked 21st).

## 4. Discussion

### 4.1. The Development of Urbanized Rural Areas

Some researchers have found that rural urbanization is an effective approach to confront rural marginalization and suggested that it should be encouraged in other rural areas [23,43,44]. However, rural urbanization may not be a prevalent phenomenon because, as our results regarding our first research question suggest, there are only a small number of urbanized rural areas in Fujian Province. To answer our second question, we investigated the spatial patterns of the urbanized rural areas. First, spatial clustering of urbanized rural areas is an effective strategy to compete or collaborate with urban areas and brings important benefits, such as information sharing, reductions in costs, and efficiency gains, thereby inhibiting rural marginalization [45]. Second, proximity to well-developed city centers facilitates rural urbanization. Well-developed cities (e.g., Quanzhou, Xiamen, Zhangzhou, and Fuzhou City) had larger numbers of urbanized rural areas than poorly developed cities (e.g., Longyan, Nanping, Sanming, and Ningde City). On the one hand, since the reform and open-door policies in 1978, villagers in rural areas of the well-developed cities in Fujian Province (e.g., Quanzhou City) have spontaneously organized rural enterprises that can absorb rural surplus labor and increase villagers' incomes [13]. On the other hand, urban areas in the well-developed cities have experienced rapid urbanization and industrialization and attracted more investment than other cities [38]. The developments in both urban and rural areas have contributed to rural urbanization. In contrast, the rural areas in Ningde, Sanming, and Nanping City are traditionally agricultural regions and urban areas in these cities lack industrialization and investment, which has led to rural marginalization [37]. Therefore, rural areas have more chances to undergo urbanization if they are close to well-developed city centers, which can provide more opportunities for the surrounding rural areas than less-developed city centers. Our findings are consistent with other research demonstrating that the presence of cities of considerable size is a prerequisite for rural areas to escape the fate of marginalization [45,46]. Lastly, transportation accessibility benefits rural urbanization. Rural urbanization is closely related to

labor-intensive and low-technology industries in Fujian Province, such as textiles, shoes, food and drink, and crafts [39]. The flow of products and labor in these industries relies on transportation accessibility.

*4.2. Socioeconomic Activities in Urbanized Rural Areas*

To answer our third question, we investigated the socioeconomic activities in the urbanized rural areas in Fujian Province. Our results indicate that the urbanized rural areas can be considered rudimentary urban areas in terms of socioeconomic activities. Specifically, the urbanized rural areas demonstrate an urban-like housing model, diverse non-agricultural activities, transportation improvements, and sufficient health services.

### 4.2.1. Urban-like Housing Model

Housing demand is boosted by large population sizes in urbanized rural areas. A higher average frequency of house number POIs suggests that the urbanized rural areas can provide a larger housing supply and maintain larger population sizes than the ordinary rural areas. In addition, we found that housing intensification was prevalent in the urbanized rural areas, as reflected in the residential quarter POIs in the urbanized rural areas. In China's rural housing model, villagers typically live in traditional separated family houses. The residential quarter POIs represent contemporary urban multi-story residential buildings, which are rare in rural areas. Residential quarters can accommodate much larger populations than traditional separated rural houses. Notably, the residential quarters in the urbanized rural areas were built by the local government (e.g., for displacement and resettlement) [47–49]. Villagers were forced to move from traditional rural houses to modern multi-story apartments. This phenomenon not only changes the morphology of traditional rural villages but also transforms villagers' lifestyles and livelihoods to forms that are more common in urban living environments. Thus, to accommodate large population sizes, urbanized rural areas not only provide larger housing supplies but also follow an intensified housing model.

### 4.2.2. Diverse Non-Agricultural Activities

While the Chinese rural economy is mainly agricultural, many researchers have highlighted the relevance of non-agricultural activities in rural transformation [13,24,28]. Our results likewise demonstrate the diverse non-agricultural activities in the urbanized rural areas of Fujian Province. Company POIs represent the most important non-agricultural activities in urbanized rural areas and mainly corresponded to labor-intensive and low-technology rural enterprises in Fujian Province [13,39]. The home decoration and building material shop POIs had a higher average frequency in the urbanized rural areas. We argue that this is a result of the urban-like housing model. When more houses are built and more villagers move into their new houses, the demands for home decoration materials and appliances are boosted. Other POI categories relating to non-agricultural activities (e.g., clothing and accessory shop, Chinese restaurant, convenience shop, and supermarket POIs) also indicate the diversity of non-agricultural activities in urbanized rural areas.

### 4.2.3. Transportation Improvements

In the urbanized rural areas, transportation improvements manifest in two ways: complex road networks (intersection name and street name POIs) and public transportation (bus station POIs). Most previous research has highlighted the importance of transportation between rural areas and the outside world [45,50]. We also suggest that transportation accessibility is improved within urbanized rural areas. The intersection name and street name POIs indicate that the urbanized rural areas have been transformed from one-street layouts, which are common spatial layouts in rural areas, into network-street layouts. In addition, the bus station POIs indicate that the urbanized rural areas have been integrated into the urban public transport system, which makes it more convenient for villagers to commute between their locations and urban areas.

### 4.2.4. Sufficient Health Services

Rural marginalization leads to a decline in health services in rural areas [7]. Here, we found that most rural areas in Fujian Province were extremely lacking in health services. The main providers of health services in the ordinary rural areas were township hospitals (average frequency = 0.08, ranked 16th). Township hospitals in China are public health centers that contain fewer than 100 beds and provide preventive care, minimal health care, and rehabilitation services. Besides the low capacity of these hospitals, the low average frequencies of the township hospital (0.08) and clinic (0.07) POIs indicate an uneven distribution between the demand for and supply of health services in ordinary rural areas. In contrast, the urbanized rural areas have better access to health services than the ordinary rural areas, with an average of 0.52 clinic (ranked 19th) and 0.45 pharmacy (ranked 23rd) POIs in each village-level administrative unit. Clinics in China can be opened by physicians who have been practicing medicine for at least five years after receiving a national physician license, and these clinics serve as effective supplements to rural health services. Finally, it is noteworthy that most health services in clinics and pharmacies are not covered by public insurance policies, which indicates that villagers in the urbanized rural areas can afford these health services.

### 4.3. Implications on Policy-Making and Research

Recognition that the opportunities and challenges presented by rural transformation could impede the comprehensive achievement of all the SDGs is necessary in order to relocate rural areas to the foreground of policies and research. The findings of our study have several policy implications. First, we only identified a few urbanized rural areas in Fujian Province, and our findings are an important reminder for policies that aim to generalize rural urbanization to other rural areas. These policies should channel more funds and resources to the rural areas that are capable of employing surplus labor and increasing local incomes. A generalized policy that focuses on all rural areas may waste the funds and resources that are allocated to the less capable rural areas. Second, we suggest that urbanized rural areas could play an intermediary role in regional development planning, where they could develop complementary functions upward with urban areas while also strengthening linkages downward with neighboring rural areas [45]. Third, developing diversified non-agricultural activities is the key to ensuring sustainable rural economies and societies [51]. This requires a systematic project that includes skill training, reasonable tax breakers, infrastructure improvements, regulatory environments, and so on. Lastly, steadfast government commitment to environmental protection can assist urbanized rural areas in capturing socioeconomic opportunities and in minimizing trade-offs from environmental pressures [10]. Built-up land expansion in urbanized rural areas results in farmland loss and is a great challenge for food security and farmland protection polices [39,52]. There is an urgent need to develop and implement stringent land-use policies to regulate built-up land expansion in these areas, such as primary farmland zoning and a requisition–compensation balance policy. Similarities in socioeconomic activities between urbanized rural areas and urban areas indicate that rural urbanization is associated with more urban-like energy consumption, such as high energy intensity, a high material footprint and material consumption, and high greenhouse gas emissions [53]. Urban-like energy consumption tends to damage vulnerable rural environments [13]. Furthermore, the vulnerable rural environment is a key issue in terms of the adequacy of water supplies and irrigation [54]. To mitigate the negative impacts of rural urbanization on the rural environment, local governments should develop local environmental knowledge and strengthen the role of spatial planning in rural urbanization.

Rural development represents a sustainability issue across countries. Many of the challenges and opportunities that rural areas present for sustainable development are found in the Global South. Our findings from China about spatial patterns and socioeconomic activities in urbanized rural areas are highly relevant to understanding rural urbanization in other developing countries. Our findings reinforce previous understandings that spatial

clustering [13], proximity to urban centers and main roads [9,12,14], and diverse non-agricultural activities [10,11] are vital for rural urbanization. Our study also adds new knowledge on rural urbanization. We presented the detailed socioeconomic activities in urbanized rural areas. We argue that, as well as the diverse non-agricultural activities, other socioeconomic activities (e.g., housing model, transportation, health services) deserve more attention in the future and in the context of other developing countries, as these socioeconomic activities can affect sustainable development in rural areas. Lastly, compared with previous studies, the method we used was more robust, as we included abundant samples, diverse data sources, and sufficient longitudinal information.

## 5. Conclusions

Measuring and mapping rural urbanization can effectively support policy-making toward sustainable development. To the best of our knowledge, our study may be the only attempt to measure and delimit rural urbanization at such a high granularity and with such a vast scope. Rural urbanization may not be a prevalent phenomenon, as we could only delimit a few urbanized rural areas in Fujian Province. Analysis of the spatial patterns of the urbanized rural areas showed that spatial clustering, proximity to well-developed urban centers, and transportation accessibility have contributed to rural urbanization in Fujian Province. Analysis of the socioeconomic activities in the urbanized rural areas showed that urbanized rural areas can be considered rudimentary urban areas in terms of socioeconomic activities. Specifically, we found four representative socioeconomic activities in the urbanized rural areas: an urban-like housing model, diversification of non-agricultural livelihoods, transportation improvements, and sufficient health services. Based on our findings, we suggest that rural urbanization contributes to sustainability development, such as no poverty, zero hunger, and good health and well-being. However, we advise caution in light of the negative impacts of rural urbanization on sustainable development (e.g., biodiversity loss, increasing energy intensity, land degradation).

While we are confident in the robustness of our findings, we acknowledge the following shortcomings. Firstly, we did not validate the urbanized rural areas we identified. In this study, we used urban areas as contrast samples and defined urbanized rural areas as territories where the population size, economic output, and built-up land area were larger than in urban areas. The identification standard may have been strict, meaning that we may have underestimated the number of the urbanized rural areas. Secondly, the data on GDP and population came from the raster data, which may have been relatively coarse for our study at the village level. Census data would have been an alternative source; however, such data are unavailable for the village-level administrative unit in China. Despite these limitations, we used POI data to illustrate that urbanized rural areas are similar to urban areas in terms of socioeconomic activities, which may validate our identification strategy to some extent. We believe that our findings offer a rare glimpse into rural urbanization and recommend wider application of the approach used in our study in other rural areas, especially in developing countries.

**Supplementary Materials:** The following supporting information can be downloaded at: https://www.mdpi.com/article/10.3390/land11070969/s1, Table S1: Explanation of the top 25 POI categories and related socioeconomic activities mentioned in this study.

**Author Contributions:** Conceptualization, Q.G. and Z.H.; methodology, Z.H.; software, Z.H.; validation, Q.G., Z.H., D.L. and M.S.; formal analysis, Z.H.; resources, Q.G. and Z.H.; data curation, Z.H.; writing—original draft preparation, Q.G., Z.H., D.L. and M.S.; writing—review and editing, Q.G., Z.H., D.L. and M.S.; visualization, Q.G., Z.H., D.L. and M.S.; supervision, Q.G. and M.S.; funding acquisition, Q.G. All authors have read and agreed to the published version of the manuscript.

**Funding:** This research was funded by the National Natural Science Foundation of China (Grant No. 31872688), the National Key R&D Program of China (Grant No. 2017YFC0505803), the Natural Science Foundation of Guangdong Province, China (Grant No. 2020A1515011223), and the Fundamental Research Funds for Central Universities (Grant No. 19lgpy51).

**Institutional Review Board Statement:** Not applicable.

**Informed Consent Statement:** Not applicable.

**Data Availability Statement:** The data presented in this study are available on request from the corresponding author.

**Acknowledgments:** Zhichao He would like to express his gratitude to the China Scholarship Council for supporting his PhD study. Furthermore, the authors thank Melissa Dawes for proofreading the manuscript and Anna Hersperger and Chunhong Zhao for their constructive comments and suggestions.

**Conflicts of Interest:** The authors declare no conflict of interest.

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
