# Peer review of "Analysis of Spatial Patterns and Socioeconomic Activities of Urbanized Rural Areas in Fujian Province, China"

_land, doi:10.3390/land11070969_

Round 1
Reviewer 1 Report
It is a good and well preented article. My recomendation is that the authors should work in a more precise concept of "urbanized rural areas" which is a contentious arena of the academic debate.
In my view it could be accepted in present form.
Reviewer 2 Report
Thank you for review opportunity paper titled How rural urbanization impacts sustainable development? Analysis of spatial patterns and socioeconomic activities of urbanized rural areas in Fujian Province, China. The paper is very interesting and well written. I have some suggestions to improve the paper:
- In theoretical background authors should mention that urbanized rural areas are often located around large and medium cities, where suburbanization proces causes change rural areas to urbanized areas. Some references about this type of rural urbanization should be added like:
https://doi.org/10.1016/0094-1190(84)90019-6
DOI 10.2478/bog-2013-0005
https://doi.org/10.1111/j.1468-0467.2007.00244.x
- Land is the international Journal, so I suggest to change local currency to USD. It will be more clear for non-Chinese readers.
Good luck with your paper!
Reviewer 3 Report
This is a useful and interesting paper. Perhaps , some mention could be made of the impact of urbanization on the food supply chain and also on the various rural communities, the impact on the water supply and food.

Reviewer 4 Report
The paper deals with spatial patterns and socioeconomic activities of urbanized rural areas in Fujian Province in China, matching the readership of the journal. It is a valuable contribution to the scientific knowledge because this study has defined urbanized rural areas as territories where the population size, economic output, and built-up land area were larger than in urban areas. The case study analysis is well written, figures and tables are ok.
Nevertheless, I found two main issues in the first part of the text.
In the introduction I lack a more in-depth focus on the literature review to understand the difference between the studies released on some key themes or urban-rural areas.
Another task to do is to provide some reflection on some themes that accordig to you the literature should be explored in China based on international references. Here a number of example:
- walkability and regeneration of the existing environment: https://www.mdpi.com/2071-1050/14/1/457
- Rural Transportation Infrastructure, https://www.mdpi.com/2071-1050/14/4/2149/htm
- regeneration of historic sites in rural areas, https://www.mdpi.com/2071-1050/13/9/5069
- return to villages after the pandemic outbreak? https://onlinelibrary.wiley.com/doi/full/10.1111/apv.12340
- landscape issues, https://link.springer.com/article/10.1007/s11355-021-00467-6
- gentrification, https://uknowledge.uky.edu/cgi/viewcontent.cgi?article=5464&context=klj
This is why I require a new version of the paper to check whether the required improvements are enough for publication.
Round 2
Reviewer 4 Report
Yes, to me it's fine and the manuscript is accepted.